# Leveraging Historical Interactions for Factual Review Generation with Large Language Models

## Abstract

User reviews play a crucial role in influencing purchase decisions and enhancing recommendation systems. Automatically generating high-quality reviews helps complement existing feedback by providing additional perspectives and uncovering overlooked details for consumers. LLMs have become ideal tools for this task due to their strong text generation capabilities. However, existing LLM-based methods often fail to effectively incorporate user- and item-specific interactions, limiting their ability to generate factually consistent and contextually relevant reviews. To address these challenges, we propose DyGRevLLM, an innovative framework that integrates dynamic graph representation learning with LLM-based text generation. DyGRevLLM encodes and updates user and item embeddings through a pretraining procedure designed to predict future review embeddings, aligning historical interaction data with LLM input formats. By dynamically aggregating user-item interaction information and incorporating temporal behaviors, the framework generates reviews that are factually accurate. Extensive experiments on real-world datasets demonstrate that DyGRevLLM improves the factual consistency and relevance of generated reviews while maintaining coherence. Furthermore, our proposed evaluation metrics validate the effectiveness of the framework in overcoming existing limitations, offering a better solution for dynamic personalized review generation.

## 1 Introduction

Review generation is a key task in natural language generation, as high-quality reviews enhance user feedback, personalization, and interpretability in recommendation systems (Chen et al., 2015; Shuai et al., 2022; Bittencourt et al., 2023). The emergence of large language models (LLMs)(Radford et al., 2019; Brown et al., 2020) has improved fluency and diversity in text generation, surpassing traditional deep learning methods(Devlin, 2018), making LLMs ideal candidates for review generation. Yet, effective reviews depend not only on textual fluency, but also on the reviewing user, the reviewed item, and the review time(Li et al., 2021; 2023; Xie et al., 2023). Without such contextual information, LLMs may generate content misaligned with actual user needs or item attributes.

Before LLMs, deep learning approaches modeled user preferences and item features to achieve personalized review generation (Li et al., 2023; Xie et al., 2023), typically relying on historical user-item interactions. However, these interactions evolve over time—user intents shift, and item perceptions change—affecting review semantics, as illustrated in Figure 1(a). While deep models can partially capture such dynamics (Zhang et al., 2022; Tang et al., 2023), LLMs primarily process explicit textual inputs (Pan et al., 2024; Grinsztajn et al., 2023), making it unclear how to incorporate non-textual, time-evolving signals into review generation. This gap hinders LLMs from adapting to evolving behavioral patterns, limiting their ability to reflect nuanced user preferences or item-specific context over time.

Our goal is to fully harness the generative strength of LLMs while enabling them to understand and utilize dynamic user–item interactions to produce factually consistent reviews. As shown in Figure 1(b), this involves three key challenges: *(1) How to capture and incorporate dynamic historical interaction signals.* User–item interactions contain rich and evolving information—such as

preferences, attributes, and feedback—that shape review content. Effective generation requires dynamically aggregating this information to extract features that serve as conditional inputs. *(2) How to make LLMs understand non-textual interaction features.* While LLMs excel at processing text, user and item features and their implicit dynamics are not directly accessible to them. A key challenge is mapping these signals into semantic representations interpretable by LLMs. *(3) How to evaluate factual quality of generated reviews.* Traditional metrics like BLEU and ROUGE emphasize surface similarity but overlook factual consistency. Accurate evaluation demands new metrics that assess alignment with user-item context and personalization.

To tackle the above challenges, we propose DyGRevLLM, a review generation framework that integrates dynamic graph representation learning with the generative strength of large language models. It dynamically aggregates user and item traits along with their interactions to generate factually consistent and personalized reviews. Specifically, DyGRevLLM encodes users, items, timestamps, and historical reviews into embeddings, modeling their relationships from historical interactions to capture evolving preferences and item features. A dynamic graph-based pretraining task updates user/item states and predicts future review representations, aligning interaction context with the semantic space of LLMs. These predicted representations are then combined with historical reviews via prompts to guide the LLM, ensuring contextual relevance and factual accuracy. We further design two factual evaluation metrics—Descriptive Recall Alignment (DRA) and Descriptive Jaccard Consistency (DJC)—to assess alignment with historical user–item context. Experiments on real-world datasets from Amazon and Yelp show that DyGRevLLM outperforms prior LLM-based methods in review quality, factual consistency, and personalization. The constructed future review representations help LLMs better reflect users' current preferences and behavior. The main contributions of this paper are as follows:

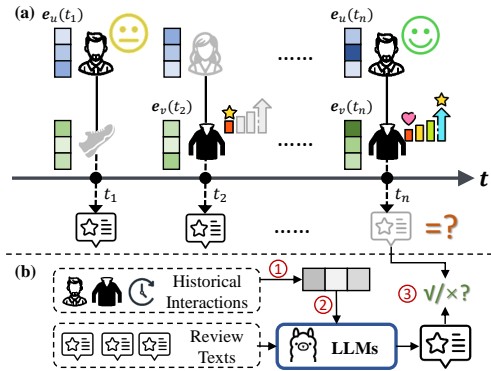

Figure 1: Illustration of LLM-based Review Generation. (a) The dynamic evolution of user and item characteristics over time and interactions affects review content, posing (b) three key challenges for LLM-based review generation: modeling historical interactions, enabling LLMs to understand them, and evaluating the factual consistency of generated reviews.

- We propose an innovative LLM-based review generation framework, DyGRevLLM, which effectively incorporates the language generation capabilities of LLMs with dynamic graph representation learning methods. The framework leverages historical user-item interaction data to improve the factual consistency of generated reviews.

- We design a pretraining task that leverages the historical interactions between users and items to predict future review representations. By aggregating review-related information and encoding the dynamic context of historical interactions, the constructed review representations serve as conditional information to guide the LLM in generating factually consistent reviews.

- Extensive experiments demonstrate that DyGRevLLM outperforms existing LLM-based methods in generating factual reviews. The proposed factual evaluation metrics and case analysis reflect that DyGRevLLM effectively mitigates factual inconsistencies in the generated reviews.

## 2 PRELIMINARIES

**Review Interaction.** We consider the recommendation system as a bipartite dynamic graph system, where the node types consist of two categories: users $U$ and items $V$. The action of a user posting a review is treated as an interaction in the dynamic graph. A valid review interaction is defined as a quadruple $s = (u, v, r, t)$, where $u \in U$ represents the user, $v \in V$ represents the item, $r$ denotes the review content, and $t$ is the timestamp of the interaction.

**Review.** In this work, all reviews are in textual format. For a more rigorous formalization, a review is denoted as $r^t_{(u,v)}$ , indicating that the review is posted by user $u$ for item $v$ at time $t$.

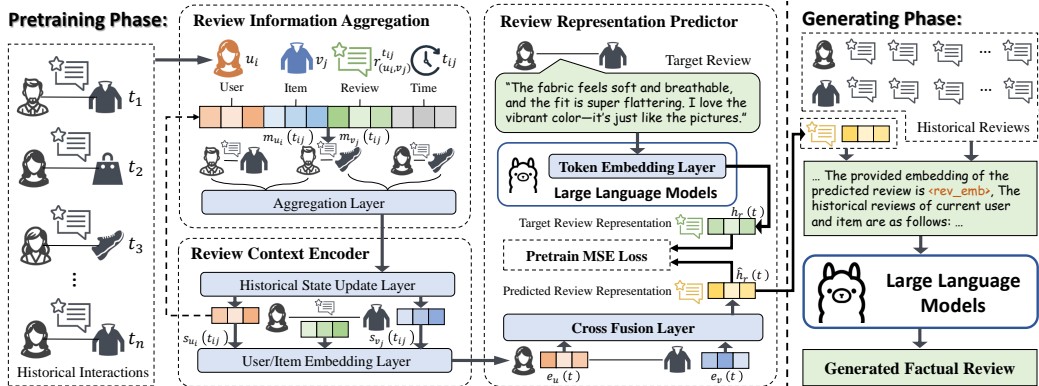

Figure 2: The overview of DyGRevLLM. DyGRevLLM encodes and aggregates historical user, item, and review information, constructs future review representations as conditional information through a pretraining task, and combines historical reviews with predicted review representations as input to the LLM for generating factual reviews.

**Problem Definition (Review Generation).** Given a user $u$ and an item $v$, the task is to generate the review content of $u$ for $v$ at the next timestamp $t$ by large language models. The generation process utilizes the historical interaction information provided by the pre-trained model $\gamma$, as well as the historical review sequences $S_u^{t-}$ and $S_v^{t-}$, where $S_u^{t-} = \{r_{(u,v_i)}^{t_i}|v_i \in V, t_i < t\}$ and $S_v^{t-} = \{r_{(u_j,v)}^{t_j}|u_j \in U, t_j < t\}$. Formally, the task can be expressed as:

$$\hat{r}_{(u,v)}^t = \Phi_{\text{LLM}}\left(\gamma(u,v), S_u^{t-}, S_v^{t-}\right) \tag{1}$$

## 3 METHODOLOGY

As illustrated in Figure 2, we propose DyGRevLLM, an LLM-based review generation framework that integrates dynamic graph representation learning with the generative capabilities of LLMs. It aggregates user and item characteristics with their historical interactions to generate factual and personalized reviews. Specifically, DyGRevLLM first encodes and aggregates user-item-review relations from historical interactions, then employs a pretraining task as an objective to update user and item states by predicting future review representations, aligning them with LLM input semantics. Finally, the model combines historical reviews with predicted representations as LLM prompts to generate the target review.

### 3.1 PRETRAINING FOR REVIEW INFORMATION

While LLMs possess strong generative capabilities, existing LLM-based review generation methods often fail to incorporate and interpret the dynamic nature of historical user-item interactions. To address this, we introduce a pretraining phase that encodes historical interactions into future review representations, serving as conditional guidance for the LLM during generation. This phase includes three components: Review Information Aggregation, Review Context Encoder, and Future Review Representation Predictor.

#### 3.1.1 REVIEW INFORMATION AGGREGATION

To provide historical context for future review generation, we propose the Review Information Aggregation module, which encodes and aggregates key interaction elements, including review content, timestamps, and user and item characteristics. Specifically, we first encode review-related information. Inspired by Rossi et al. (2020), we maintain a separate embedding for each user and item, denoted as $\boldsymbol{s}_{u_i}(t_{ij}^-)$ and $\boldsymbol{s}_{v_j}(t_{ij}^-)$, representing the historical state of user $u_i$ and item $v_j$ before time $t_{ij}$. Taking user $u_i$ as an example, for a given interaction $(u_i, v_j, r_{(u_i,v_j)}^{t_{ij}}, t_{ij})$ between user $u_i$ and

item $v_j$ at time $t_{ij}$, the corresponding review information is encoded as:

$$\boldsymbol{m}_{u_i}(t_{ij}) = \sigma(\boldsymbol{W}_m[\boldsymbol{s}_{u_i}(t_{ij}^-)\|\boldsymbol{s}_{v_j}(t_{ij}^-)\|f_r(r_{(u_i,v_j)}^{t_{ij}})\|f_t(t_{ij})] + \boldsymbol{b}_m), \tag{2}$$

where $\|$ denotes the concatenation operation, $\boldsymbol{W}_m$ and $\boldsymbol{b}_m$ are trainable parameters, and $\sigma$ is the activation function. The function $f_r(\cdot)$ encodes the review content using the token embedding layer of the open-source LLM, while $f_t(\cdot)$ encodes temporal information using a periodic function (e.g., sine and cosine functions) (Vaswani, 2017; Rossi et al., 2020) to capture time-dependent dynamics.

Then, we aggregate the encoded interaction information according to users and items to model the dynamic evolution of their historical interactions. For example, for user $u_i$, given all historical interactions before time $t_{ij}$ in the set $\{(u_i, v_j, r_{ij}, t_k)|v_j \in \mathcal{N}(u_i), t_k < t_{ij}\}$, where $\mathcal{N}(u_i)$ demotes the neighbors of $u_i$, the review information encodings are aggregated as follows:

$$\boldsymbol{c}_{u_i}(t_{ij}) = \text{Agg}(\boldsymbol{m}_{u_i}(t_1), \dots, \boldsymbol{m}_{u_i}(t_k)), \tag{3}$$

where $\boldsymbol{c}_{u_i}(t_{ij})$ denotes the aggregated review information, and the aggregation function $\text{Agg}(\cdot)$ is implemented by a single linear layer. Through this approach, we aggregate the review content and corresponding historical interaction information into the user and item embeddings $\boldsymbol{c}_{u_i}(t_{ij})$ and $\boldsymbol{c}_{v_j}(t_{ij})$. These embeddings serve as critical inputs for the subsequent review dynamic context encoding, providing rich historical contextual semantics to support the construction of future review representations.

### 3.1.2 Review Context Encoder

To model the evolving states of users and items during review interactions, we design the Review Context Encoder, which updates user/item embeddings via historical information and extracts key contextual features through a time-aware attention mechanism. Specifically, for a user $u_i$ and item $v_j$ interaction at time $t_{ij}$, we use the aggregated review information $\boldsymbol{c}_{u_i}(t_{ij})$ and $\boldsymbol{c}_{v_j}(t_{ij})$ to update historical state embeddings:

$$\begin{aligned}\boldsymbol{s}_{u_i}(t_{ij}) &= f_{upt}\left(\boldsymbol{s}_{u_i}(t_{ij}^-), \boldsymbol{c}_{u_i}(t_{ij})\right), \\ \boldsymbol{s}_{v_j}(t_{ij}) &= f_{upt}\left(\boldsymbol{s}_{v_j}(t_{ij}^-), \boldsymbol{c}_{v_j}(t_{ij})\right),\end{aligned} \tag{4}$$

where $f_{upd}(\cdot)$ is the state update function, implemented by a gated recurrent unit (GRU) in this work to fuse the historical state with the current interaction features. The updated state embeddings $\boldsymbol{s}_{u_i}(t_{ij})$ and $\boldsymbol{s}_{v_j}(t_{ij})$ reflect the evolving characteristics of users and items at the current time.

Then, to select the most informative historical interactions and generate the immediate user and item representations $\boldsymbol{e}_u(t)$ and $\boldsymbol{e}_v(t)$, we employ a time-aware attention mechanism (Xu et al., 2023). Taking the user node $u_i$ as an example, let the hidden state of the user at layer $l$ be denoted as $\boldsymbol{h}_i^{(l)}(t)$, with the initial state $h_i^{(0)}(t) = \boldsymbol{s}_{u_i}(t)$. We focus on the most recent $K$ interactions involving the user $u_i$ and its interacted items, compute the attention weight $\alpha_k$, and aggregate the historical information as follows:

$$\begin{aligned}\boldsymbol{e}_u(t) &= \boldsymbol{h}_i^{(L)}(t) = \text{MLP}\left(h_i^{(L-1)}(t)\|\Sigma_{j \in \mathcal{N}(u_i)}\alpha_{ij} \cdot \boldsymbol{z}_{ij}(t)\right), \\ \boldsymbol{z}_{ij}(t) &= \boldsymbol{W}_z\left[\boldsymbol{h}_j^{(L-1)}(t)\|f_r(r_{(u_i,v_j)}^t)\|f_t(t)\right],\end{aligned} \tag{5}$$

where $\boldsymbol{W}_z$ is a trainable parameter matrix. $\boldsymbol{z}_{ij}(t)$ encodes information from neighboring item nodes, considering the historical state, review content, and temporal features. The attention score $\alpha_{ij}$ is computed as:

$$\alpha_{ij} = \text{Softmax}\left((\boldsymbol{W}_q h_i^{(L-1)}(t)) \cdot (\boldsymbol{W}_k z_{ij}(t))^\top\right), \tag{6}$$

where $\boldsymbol{W}_q$ and $\boldsymbol{W}_k$ are trainable projection matrices. The computation of the item representation $\boldsymbol{e}_v(t)$ follows the same procedure as the user representation. Through this process, the pretraining model focuses on temporally relevant and contextually important historical states, leading to more accurate user and item embeddings. The final user embedding $\boldsymbol{e}_u(t)$ and item embedding $\boldsymbol{e}_v(t)$ incorporate review context and historical interaction information and serve as inputs for future review representation prediction.

### 3.1.3 FUTURE REVIEW REPRESENTATION PREDICTOR

To bridge user-item interaction features with the semantic space of LLMs, we design a Future Review Representation Predictor, which generates a review embedding conditioned on historical user and item states. Given the user embedding $e_u(t)$ and item embedding $ev(t)$, we first design a Cross Fusion Layer: both embeddings are projected into a unified space, then concatenated and passed through a multi-head attention to obtain the predicted review representation:

$$\hat{h}_r(t) = \text{MLP}\left(\text{MHA}(W_u e_u(t)) \| W_v e_v(t))\right), \tag{7}$$

where $W_u$ and $W_v$ are trainable parameters, and $\text{MHA}(\cdot)$ stands for multi-head self attention. The output $\hat{h}_r(t)$ represents the predicted future review representation.

Then, to guide the pretraining model to construct the review representation consistent with the actual review, we extract the ground truth by passing the target review text $r_{(u,v)}^t$ through the first token embedding layer of the LLM:

$$h_r(t) = \text{TokenEmbedding}\left(r_{(u,v)}^t\right), \tag{8}$$

where $h_r(t)$ denotes the ground truth of the target review representation. The optimization objective of the pretraining model is to compute the Mean Squared Error (MSE) loss between the predicted review representation $\hat{h}_r(t)$ and the target review representation $h_r(t)$, which is formalized as:

$$\mathcal{L}(\hat{h}, h) = \frac{1}{d_r} \|\hat{h}_r(t) - h_r(t)\|^2, \tag{9}$$

where $d_r$ denotes the dimension of review representation. By minimizing the loss function, the pretraining model constructs future review representations that capture historical semantics and align with LLM input, serving as conditional guidance for factual review generation.

### 3.2 GENERATION FOR FACTUAL REVIEWS

Given the powerful contextual reasoning and generative capabilities of LLMs, we posit that they can effectively handle the aspects of similarity and diversity in review generation, while the factual consistency of the generated reviews is supported by the review representations provided by the pretraining model. To make the generated reviews reflect the historical interaction context and the personalized needs of users and items, we incorporate sampled historical reviews and the predicted future review representation through prompts, providing dynamic contextual support for the generation process.

Specifically, for a given user $u$ and item $v$, we first sample the most recent $k$ relevant reviews from their respective historical review sequences, denoted as $S_u^{t-} = \{r_{(u,v_i)}^{t_i} | v_i \in V, t_i < t\}$ and $S_v^{t-} = \{r_{(u_j,v)}^{t_j} | u_j \in U, t_j < t\}$. These historical reviews contain linguistic patterns and semantic features from the user's past preferences and the item's characteristics, which contribute to the generated review's similarity and contextual relevance. The predicted future review representation $\hat{h}_r(t)$ serves as the core conditional information. It is combined with the sampled historical reviews through a prompt template, forming the input to the LLM. The user and item historical reviews are presented in natural language, while the predicted review representation $\hat{h}_r(t)$ is embedded as a token $<\text{rev\_emb}>$ within the generation sequence. This design allows the LLM to integrate both past interactions and future semantic goals during the generation phase. Examples of the generation phase are provided in Appendix A.5. The generation of factual reviews can be formalized as:

$$\hat{r}_{(u,v)}^t = \Phi_{\text{LLM}}\left(\text{Prompt}(S_u^{t-}, S_v^{t-}, <\text{rev\_emb} : \hat{h}_r(t)>)\right). \tag{10}$$

Unlike methods that rely only on concatenating historical reviews, our approach leverages the historical review interaction contexts provided by the pretraining model to guide the LLM. This enables the generated reviews to maintain close ties to historical details while retaining semantic flexibility. The resulting reviews achieve a balance between contextual coherence and personalized expression, avoiding redundant or irrelevant content that may conflict with user preferences or the attributes of the item.

## 4 EXPERIMENTS

In this section, we conduct extensive experiments on three real-world datasets to evaluate the effectiveness of the proposed method for future review generation. The experiments aim to address the following five research questions:

1) **RQ1:** Can the proposed method improve the quality of reviews generated by LLM compared to existing methods? 2) **RQ2:** Can the proposed modules improve the understanding of the LLM of historical review interactions? 3) **RQ3:** Is the quality of reviews generated by LLM influenced by historical reviews of users and items? 4) **RQ4:** Are future review representations effective for review generation? 5) **RQ5:** How can the factual consistency of the generated reviews be reasonably evaluated?

### 4.1 EXPERIMENTAL SETTINGS

**Datasets.** We conduct extensive experiments on three public datasets: **Amazon-Book, Amazon-Cloth**, and **YELP**, each containing users, items, reviews, and timestamps. Dataset statistics are shown in Table 1, with source and preprocessing details in Appendix A.1.

Table 1: Statistics of the preprocessed datasets.

| Dataset | Customers | Products | Reviews | Time Span |
|---|---|---|---|---|
| Amazon-Book | 4825 | 4716 | 414010 | 10 years |
| Amazon-Cloth | 5777 | 135 | 187433 | 4 years |
| YELP | 7979 | 7325 | 505998 | 11 years |

**Baselines.** To comprehensively evaluate our method, we compare it with three categories of baselines: 1) classic deep learning-based explainable recommendation methods, including Att2Seq(Dong et al., 2017) and PETER(Li et al., 2021); 2) standalone large language models, including open-source models ChatGLM-6B(GLM et al., 2024), Baichuan-7B(Yang et al., 2023), Persimmon-8B(Elsen et al., 2023), as well as the closed-source APIs ChatGPT-4o(OpenAI, 2023) and DeepSeek-r1(Guo et al., 2025); and 3) LLM-based review generation methods, including PRAG-LLM(Xie et al., 2023) and Review-LLM(Peng et al., 2024). Detailed descriptions of all baseline methods are provided in Appendix A.2. For a fair comparison, PRAG-LLM, Review-LLM, and our method are implemented using LLAMA-8B as the backbone model.

**Evaluation Metrics.** In addition to traditional text similarity metrics, we emphasize evaluating the factual and semantic consistency of generated reviews, particularly their alignment with user preferences and item characteristics. We adopt two categories of metrics: similarity metrics, including BLEU and BERTScore (BERT-S), to assess lexical and semantic overlap; and factual metrics, including two proposed metrics—Descriptive Recall Alignment (DRA) and Descriptive Jaccard Consistency (DJC). Detailed computation methods are provided in Appendix A.3.

- **Descriptive Recall Alignment (DRA)**: Measures the extent to which the generated review successfully covers the key descriptive features (e.g., important adjectives and attributes) found in the target review. Inspired by Recall, a higher DRA score indicates better semantic coverage of core factual elements.

- **Descriptive Jaccard Consistency (DJC)**: Evaluates the semantic consistency between generated and target reviews based on the overlap of descriptive elements. Derived from Jaccard Similarity, DJC emphasizes meaningful alignment of user- and item-related features rather than surface-level word matches.

**Hyperparameter settings.** Each dataset is split into pretraining and generation data, with approximately 7,000 reviews per dataset used for evaluation. The pretraining data are further divided into 70% training, 15% validation, and 15% testing. We use **LLAMA3-8B** as the backbone model for generation, and set the historical review sequence length $k = 5$ for both users and items. Additional implementation details and hyperparameters are provided in Appendix A.4.

### 4.2 OVERALL PERFORMANCE AND HUMAN EVALUATION (RQ1)

We compare DyGRevLLM with baselines from three categories: deep learning-based explainable methods, standalone LLMs, and LLM-based review generation models. Table 2 reports perfor-

Table 2: Performance comparison of the proposed method, deep learning-based recommendation methods, standalone LLMs, and LLM-based review generation methods. BERT-S denotes BERTScore, and all metric values are scaled by $10^2$.

| Model | Amazon-Book | | | | Amazon-Cloth | | | | YELP | | | |
|---|---|---|---|---|---|---|---|---|---|---|---|---|
| | DRA | DJC | BLEU | BERT-S | DRA | DJC | BLEU | BERT-S | DRA | DJC | BLEU | BERT-S |
| Att2Seq | 11.42 | 3.16 | 0.12 | 35.21 | 2.00 | 1.40 | 0.01 | 42.41 | 7.76 | 5.66 | 0.02 | 34.45 |
| PETER | 12.28 | 4.24 | 0.08 | 42.21 | 4.72 | 3.37 | 1.21 | 38.15 | 10.70 | 5.28 | 0.01 | 41.45 |
| ChatGLM-6B | 7.13 | 4.38 | 0.30 | 43.45 | 8.87 | 7.32 | 1.28 | 36.82 | 8.18 | 7.58 | 0.61 | 44.42 |
| Baichuan-7B | 6.74 | 4.23 | 0.38 | 43.62 | 7.73 | 7.66 | 2.16 | 32.09 | 7.70 | 7.09 | 0.30 | 44.05 |
| Persimmon-8B | 8.15 | 4.93 | 2.63 | 33.75 | 6.92 | 6.85 | 2.14 | 31.87 | 7.78 | 7.17 | 0.33 | 44.19 |
| ChatGPT-4o | 8.82 | 4.55 | 0.10 | 46.51 | 11.98 | 9.02 | 5.47 | 38.62 | 8.23 | 6.06 | 0.05 | 44.62 |
| Deepseek-r1 | 7.46 | 4.41 | 0.71 | 44.42 | 10.79 | 8.53 | 5.70 | 35.10 | 8.43 | 6.39 | 0.17 | 42.91 |
| PRAG-LLM | 5.91 | 3.01 | 0.10 | 30.05 | 2.93 | 2.14 | 0.03 | 33.84 | 9.78 | 4.75 | 0.05 | 46.64 |
| Review-LLM | 12.75 | 4.76 | 0.28 | 47.73 | 12.23 | 9.03 | 2.53 | 44.04 | 15.77 | 6.52 | 0.13 | 48.45 |
| DyGRevLLM | 13.12 | 5.08 | 0.38 | 46.78 | 16.58 | 12.58 | 3.58 | 43.12 | 16.31 | 7.83 | 0.30 | 49.28 |

Table 3: Average human ranking of generated reviews across 100 samples per dataset.

| Dataset | Att2Seq | PETER | ChatGLM | Baichuan | Persimmon | ChatGPT | DeepSeek | PRAG-LLM | Review-LLM | DyGRevLLM |
|---|---|---|---|---|---|---|---|---|---|---|
| Amazon-Book | 4.68 | 4.11 | 6.57 | 8.16 | 8.16 | 2.58 | 3.87 | 8.49 | 6.30 | 1.96 |
| Amazon-Cloth | 4.43 | 6.55 | 8.13 | 8.33 | 8.93 | 2.60 | 3.89 | 8.66 | 6.35 | 2.03 |
| YELP | 6.14 | 4.64 | 6.57 | 8.20 | 8.26 | 3.92 | 3.99 | 8.55 | 2.66 | 1.94 |

mance across three datasets using both factual metrics (DRA, DJC) and similarity metrics (BLEU, BERTScore).

DyGRevLLM achieves the best DRA and DJC scores across all datasets, demonstrating superior factual consistency. Compared with Att2Seq and PETER, it shows significant gains in semantic alignment. Among standalone LLMs, open-source models (e.g., ChatGLM, Baichuan) and API-based models (e.g., ChatGPT-4o, DeepSeek-r1) perform well on fluency but fall short in factual accuracy. Review-LLM benefits from prompt tuning but still underperforms our method, which explicitly integrates historical user-item interactions. DyGRevLLM also achieves competitive BLEU and BERTScore results, ensuring overall review quality.

To further assess review quality, we conduct a human evaluation by asking 30 annotators to rank model outputs for 100 samples per dataset. As shown in Table 3, DyGRevLLM consistently ranks first, followed by ChatGPT-4o and Review-LLM. These results not only confirm the effectiveness of our model but also validate the proposed factual metrics: DRA and DJC show strong alignment with human judgment, supporting their reliability in evaluating review consistency and informativeness.

### 4.3 ABLATION STUDY (RQ2)

We conduct ablation studies to evaluate the contribution of key components in DyGRevLLM, including: (1) removing the future review representation (*w/o CondInfo*); (2) excluding item embeddings during pretraining (*w/o item*); and (3) excluding user embeddings during pretraining (*w/o user*). Results on all three datasets are shown in Figure 3, evaluated using DRA and DJC as the primary factual consistency metrics.

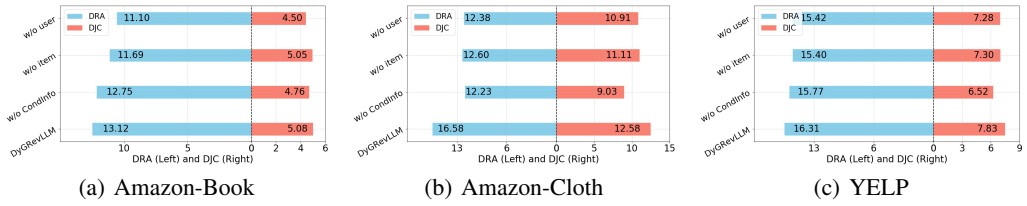

(a) Amazon-Book       (b) Amazon-Cloth       (c) YELP

Figure 3: Ablation study of DRA and DJC metrics on three datasets.

The *w/o CondInfo* variant demonstrates inferior performance across all three datasets in terms of factual consistency, confirming the necessity of future review representations in capturing factual and dynamic context. Both *w/o item* and *w/o user* show moderate drops, indicating that user and

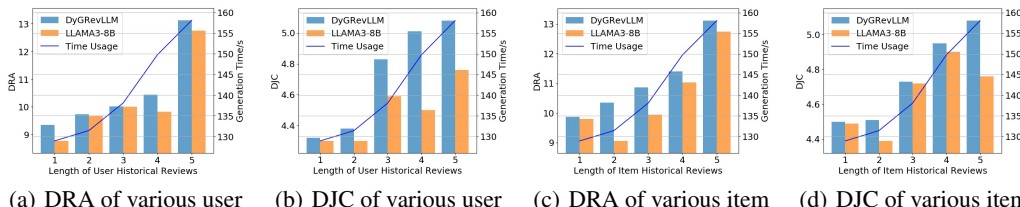

(a) DRA of various user   (b) DJC of various user   (c) DRA of various item   (d) DJC of various item

Figure 4: Sensitivity analysis of user/item historical review sequence length parameter on Amazon-Book (in terms of DRA and DJC).

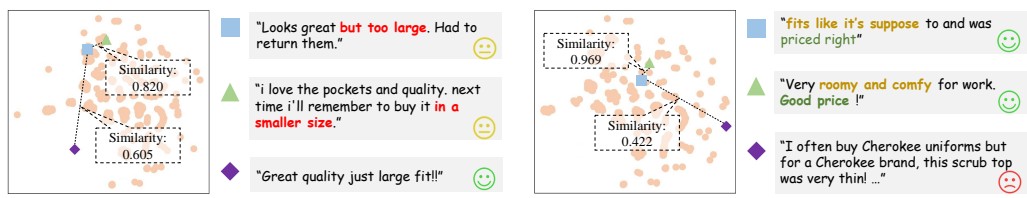

Figure 5: Visualization of future review representations with corresponding review texts.

item features are complementary and jointly essential for factual consistency. DyGRevLLM consistently outperforms all ablations, demonstrating the effectiveness of incorporating both historical interactions and structured representations for guiding LLM-based review generation.

## 4.4 PARAMETER SENSITIVITY ANALYSIS (RQ3)

We analyze how the length of historical review sequences affects review quality and efficiency. Using the Amazon-Book dataset, we compare DyGRevLLM with its backbone model (LLAMA3-8B) by varying the number of user or item historical reviews k from 1 to 5, while fixing the other to 5. We report DRA, DJC, and generation time in Figure 4.

Both models show sensitivity to the sequence length, confirming the importance of historical context from both users and items. As k increases, DyGRevLLM shows growing advantages over the backbone in factual consistency, especially on the DJC metric. However, longer sequences incur higher generation time. To balance quality and efficiency, we set k=5 as the default in our experiments.

## 4.5 VISUALIZATION OF FUTURE REVIEW REPRESENTATION (RQ4)

To validate the semantic quality of future review representations, we visualize a batch of samples using t-SNE and compare their representations with the corresponding texts. As shown in Figure 5, we highlight review pairs with high (square-triangle) and low (square-diamond) cosine similarities to illustrate semantic alignment. The visualizations show that the review representations exhibit a clustered distribution, where reviews with similar textual content have higher representation similarities, indicating that our learned representations can effectively distinguish between different review contents and map them into the LLM's semantic space. This confirms their utility as conditional inputs that help LLMs generate factually consistent and contextually grounded reviews.

## 4.6 FACTUAL CONSISTENCY ANALYSIS (RQ5)

To illustrate the advantages of our method in review generation, we compare reviews generated by DyGRevLLM and the standalone LLM (LLAMA3-8B) against ground truth. As shown in Figure 6, two representative cases are selected with color highlights indicating semantically similar content. In the first case, both DyGRevLLM and the standalone LLM generated basic factual descriptions of the item, likely derived from common phrases in historical user reviews, such as "the price is reasonable." However, DyGRevLLM captures user- and item-specific details, reflecting the benefit of incorporating historical embeddings. In the second case, where the ground truth is brief, DyGRevLLM aligns more closely with the user's current intent, while the standalone LLM introduces

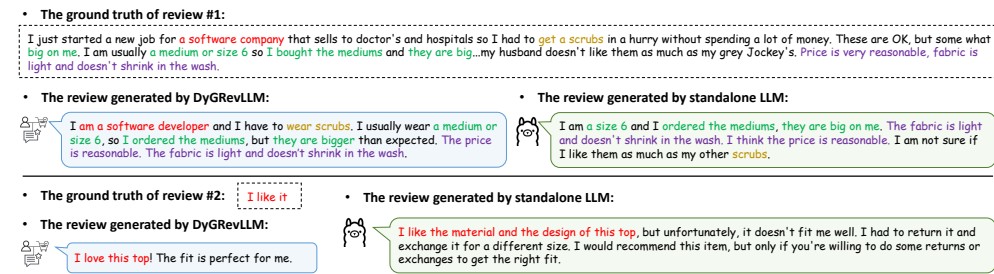

Figure 6: Examples comparing reviews generated by DyGRevLLM and standalone LLM with ground truth reviews.

redundant content from irrelevant history. These cases demonstrate that DyGRevLLM effectively integrates user, item, time, and interactions to produce reviews that are both factually accurate and contextually appropriate.

## 5 RELATED WORKS

Large language models (LLMs) have demonstrated strong reasoning and generation capabilities in diverse NLP tasks, such as summarization, machine translation, and QA (Betrián & Kaltenbrunner, 2024; Zhu et al., 2023; Luo Lab, 2023; Brown et al., 2020; Chowdhery et al., 2023; OpenAI, 2023; Raffel et al., 2020; Radford et al., 2019). Review generation, as a domain-specific task, requires not only fluency and informativeness (Li et al., 2021; 2020), but also factual consistency (Geng et al., 2022). Earlier deep learning methods were limited in generation quality, but incorporated user/item knowledge to enhance factual alignment (Hada & Shevade, 2021; Li et al., 2023; Xie et al., 2023). For instance, Xie et al. (Xie et al., 2023) mitigated hallucination through keyword extraction and templates. More recently, LLM-based approaches attempt to generate personalized reviews by prompting with product titles and historical reviews (Peng et al., 2024), yet they typically ignore the dynamic nature of user-item interactions and their evolving correlations.

Historical interactions have long been explored in recommender systems (Zhang et al., 2022; Tang et al., 2023; Ye et al., 2021), especially for sequential recommendations, where review signals are often used to aid explainability (Cheng et al., 2019; Pan et al., 2022). To capture dynamic patterns in interactions, we adopt dynamic graph representation learning (Kumar et al., 2019), widely used in link prediction tasks (Rossi et al., 2020; Cong et al., 2022; Yu et al., 2023), where updating user/item embeddings improves accuracy (Zhang et al., 2022; Wang et al., 2021). Our pretraining design is inspired by memory-based dynamic graph methods (Kumar et al., 2019; Rossi et al., 2020), which enable continual updates via past interactions. To bridge the modality gap, multimodal LLM research (Radford et al., 2021; Tan et al., 2024; Moiseev et al., 2022) offers insight on embedding external knowledge into LLMs, and has shown success across domains including mathematics (Golkar et al., 2023), vision (Radford et al., 2021), time series (Sun et al., 2023), and urban computing (Li et al., 2024). Our work integrates LLM generation with dynamic graph learning to model evolving interaction features for generating factually consistent reviews.

## 6 CONCLUSION

We proposed DyGRevLLM, a framework for generating factually consistent reviews, combining the text generation capabilities of LLMs with dynamic graph learning. By aggregating the dynamic historical interactions between users and items during the pretraining phase, the model constructs future review representations based on the historical states of users and items, guiding the LLM to generate informative reviews that better reflect the factual information. Experimental results demonstrate that our method outperforms existing LLM-based review generation approaches in terms of review quality. Furthermore, the proposed factual evaluation metrics effectively assess whether the reviews generated have factual consistency. In the future, we will improve the interpretability of the review generation process by revealing the specific contributions of historical interactions to the review generation process, improving trust in practical scenarios and real-world deployments.

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

# A APPENDIX

In this section, we provide additional details on the experimental settings, including the datasets and the proposed factual evaluation metrics. We also present an example of review generation using the proposed method. Our code is available at this link[1].

## A.1 DETAILS FOR DATASETS

The datasets used in this work are sourced from two e-commerce platforms, Amazon and YELP, comprising a total of three public available datasets. The selected datasets meet the requirement of containing four essential fields: the reviewer (user), the reviewed entity (item), the review text, and the review timestamp. The three datasets used are as follows:

---

[1] https://anonymous.4open.science/r/DyGRevLlama/

- **Amazon-Book** [2]: A dataset containing customer reviews, ratings, and timestamps for books.
- **Amazon-Cloth** [3]: A dataset containing customer feedback, ratings, and timestamps for clothing items.
- **Yelp** [4]: A dataset consisting of customer reviews, ratings, and timestamps for businesses, primarily in the dining and service industries.

To support the requirements of the proposed method during the pretraining phase, we preprocess the datasets to construct complete and informative datasets suitable for training with dynamic graph representation learning methods. Specifically, due to the large size of the original datasets, we selected data from the first few months of each dataset to ensure efficiency and feasibility. Furthermore, to construct a dense dynamic graph (customer-review-product), which is crucial for effective structural learning, we applied the following sampling strategy:

- **Density Preservation**: During sampling, we ensured that each node (customers or items) had a sufficient number of connected edges (reviews) to avoid sparsity issues.
- **Temporal Consistency**: By selecting data over a longer period of time, we ensure that the long-term habits of nodes (customers and items) are captured, enhancing the temporal characteristics of dynamic graphs.

Through the preprocessing and sampling process, we construct dense dynamic graphs with sufficient temporal and structural properties, ensuring a robust foundation for subsequent experiments.

### A.2 DETAILS FOR BASELINES

We provide the details of baseline methods in this section. The first category is classic deep learning-based explainable recommendation methods, including the following two methods:

- **Att2Seq** (Dong et al., 2017) is a sequence-to-sequence model incorporating attention mechanisms to generate reviews based on user and item embeddings. It serves as a representative method in early personalized review generation.
- **PETER** (Li et al., 2021) enhances review generation by injecting attribute-level information from users and items through a template-based decoder, improving the explainability and controllability of generated text.

The second category comprises standalone large language models, including the following five methods:

- **ChatGLM-6B** (GLM et al., 2024) is an open dialogue model based on the general language model architecture, optimized for conversational scenarios.
- **Baichuan-7B** (Yang et al., 2023) is an open-source, large-scale pre-trained language model containing 7 billion parameters.
- **Persimmon-8B** (Elsen et al., 2023) is a fully permissively licensed language model with 8 billion parameters, designed to be efficient and accessible for a wide range of applications.
- **ChatGPT-4o** (OpenAI, 2023) is a proprietary large-scale LLM accessible via API, demonstrating strong performance in general and personalized generation tasks.
- **DeepSeek-r1** (Guo et al., 2025) is a commercial LLM optimized for long-context understanding, accessed through a public API.

The third category is LLM-based review generation methods, including the following two methods:

- **PRAG-LLM** is a variant of PRAG (Xie et al., 2023), which utilizes BERT as its backbone. It accounts for the influence of both users and items to generate personalized reviews. We replace its backbone with LLMs for fair comparisons.

---

[2]https://mcauleylab.ucsd.edu/public_datasets/data/amazon_v2/categoryFiles/Books.json.gz

[3]https://mcauleylab.ucsd.edu/public_datasets/data/amazon_v2/categoryFiles/Clothing_Shoes_and_Jewelry.json.gz

[4]https://business.yelp.com/external-assets/files/Yelp-JSON.zip

- **Review-LLM** (Peng et al., 2024) is a review generation method employing strategies that combine prompt with supervised fine-tuning to generate reviews using LLMs.

## A.3 Details for the Proposed Factual Metrics

This section provides the detailed computation methods for the two factual metrics designed to evaluate the factual consistency and semantic alignment of generated reviews with user preferences and item-specific characteristics.

**Descriptive Recall Alignment (DRA):** DRA measures how effectively the generated review captures the key descriptive elements (e.g., adjectives and attributes) from the target review. Formally, let $A_g$ and $A_t$ represent the sets of descriptive features (adjectives or attributes) extracted from the generated review and target review, respectively. The DRA score is computed as:

$$\text{DRA} = \frac{|A_g \cap A_t|}{|A_t|}, \tag{11}$$

where $|A_g \cap A_t|$ denotes the number of overlapping descriptive features between the generated and target reviews, and $|A_t|$ is the total number of descriptive features in the target review. A higher DRA score indicates that the generated review effectively covers essential descriptive details, enhancing its semantic relevance and factual alignment.

**Descriptive Jaccard Consistency (DJC):** DJC evaluates the overall consistency by comparing the overlap of descriptive features in the generated and target reviews, ensuring that the shared descriptive information reflects meaningful semantic alignment. DJC is computed as:

$$\text{DJC} = \frac{|A_g \cap A_t|}{|A_g \cup A_t|}, \tag{12}$$

where $|A_g \cup A_t|$ denotes the union of descriptive features present in both the generated and target reviews, and $|A_g \cap A_t|$ represents the shared descriptive features between the two reviews. Unlike simple word-overlap metrics, DJC focuses on the meaningful alignment of descriptive elements, emphasizing features that reflect user feedback and item-specific characteristics. A higher DJC score indicates that the generated review is semantically consistent with the target review and effectively aligns with factual descriptions.

## A.4 Hyperparameter and Implementation Details

In our experiments, each dataset is split into two parts: pretraining data and generation data. Approximately 7,000 reviews are selected from each dataset for generation evaluation, while the remaining data are used for pretraining. The pretraining set is further divided into 70% for training, 15% for validation, and 15% for testing.

For the pretraining model, we use a batch size of 100, with a time-aware attention layer consisting of 1 attention layer and 2 attention heads. The learning rate is initialized at 0.001, and the dimension of the review representation is set to 4096. We adopt LLAMA3-8B as the backbone language model. During generation, the temperature is set to 0.7 to balance factual consistency and diversity. The length of historical user and item review sequences is fixed at $k = 5$ across all experiments.

## A.5 Examples of Review Generation

To provide a clearer illustration of our review generation process, we present an example from the YELP dataset, showcasing the prompt, historical review texts, inference process, and generated output. The historical reviews and future review representation are organized using formatted text and provided to the LLM through a prompt template. The LLM used is LLAMA3-8B, and the example is detailed in Table 4. From the generated results, we observe that the LLM effectively analyzes the user's review patterns and item-specific historical features, optimizing the generated review by integrating the future review representation.

Table 4: Examples of Review Generation

**Task Description:** You are an expert in consumer psychology and are adept at predicting what a customer's review will be when they buy an
item through historical reviews of a particular purchaser and a particular item. Remember that historical customer reviews are about other
products, and historical product reviews are from other customers. The history reviews will be give like this: customers' history reviews text:
$\{text1\}\backslash n\{text2\}\backslash n...\{textn\}\backslash n$ And historical reviews of items, it will be give like this: items' history reviews text: $\{text1\}\backslash n\{text2\}\backslash n...\{textn\}\backslash n$
**buyers' history reviews text:**
{
 Morimoto is worthy of the hype, even 8 years after its much-celebrated opening. That's practically forever in restaurant world, especially when you're talking about celeb-driven entities where style is more prized than substance. Nope, the sushi is still to die for here, as is the attentive service (big ups to John, our friendly waiter) and the minimalistic decor that reminds me of being on a Nordic submarine surrounded by penises. (Yes, you read that right. Take a look at the centerpieces on each table and you'll understand what I mean).
 The boy and I went on a recent Saturday night, having secured a last-minute reservation on Open Table for 7:45 (score!). We had actually been a few times in the past few years but moved on to other places in the city for our dining enjoyment - how do you think I write so many reviews? But since my birthday weekend was in full swing, I wanted to go to a lovely place where I could eat well without busting out of my dress. I chose Morimoto.
 It was buzzing but not packed, lively but not over the top. We were seated promptly and informed (by the affable John) that fugu was on the menu. Since we were celebrating the accomplishment of me making it through another year, it only seemed fitting that we risk our lives by eating a potentially deadly fish. It was delicious, and reminded me of a whisper or a ghost. Try it if you can.
 We went straight up sashimi / sushi in our choices and weren't disappointed. For dessert I had the chocolate pot de creme, and rejoiced in the marshmellowy texture of the creme, chocolate and caramelized bananas.
 I know it's my business to buzz around town, trying out different places. But Morimoto has proven itself to be reliable, and that goes a long way in a town built on fickle tastebuds. They've secured a place on my favorites list for sure.
}
{
 I still have pizza dreams about the pie I ate here a while back. But if you had told me I'd be all nostalgic as I was driving through the area, trying to find my way to this joint, I probably would have punched you in the nose. Let's just say you should be aware if you follow Google Maps from my house to Tacconelli's. Moving on...
 The restaurant itself is in a fine neighborhood, though, and looks like an old school, mom & pop establishment. We went with a large group, so I didn't have to be the one to reserve the dough, but I was excited and a little confused as to how a business could operate that way. They seem to be doing just great, though, so don't mind me - it was absolutely packed on the Saturday night we dined there. The pies themselves were good, the service was ok and the energy in the place was high. I like to try as many places as I can, so I'm glad I came here, but I'm not going to go out of my way to make this a regular stop for my fairly frequent pizza cravings.
}
{
 Next time you're in the mood for fancy cocktails, get thee to the lounge at Positano Coast, stat. They've just added new organic concoctions to their drink menu, and lemme tell you - they are yummy. The bartenders look like they should be wearing safety goggles and lab coats as they work - everything is precisely measured and flawlessly executed. The lounge is one of my favorite spots in the entire city to socialize with friends. Just sitting on the cozy sofas under billowing silks makes me feel like I'm on a tropical vacation. Since it's highly unlikely that I'll be taking a trip to the islands anytime soon I'll gladly make do with a festive beverage at this heavenly spot.
}
{

Despite the fact that it's usually quite crowded, I keep going back to Amada again and again. I highly suggest you order the Chef's Tasting Menu - it's like playing Russian Roulette but instead of guns and possible death there's food and happy tastebuds. Don't skip the sangria, either. It's the best I've ever had, and trust me, I've had a lot of sangria in my time.

Give yourself a full evening to enjoy your dining experience at Amada, because it's like a food orgy. The food just keeps coming and coming...was that too much information?
}
{

Chomp, chomp, snarfle, crunch, slurp.

That is my dramatic interpretation of me scarfing down one of their cannolis. Lemme tell ya - it was a life-altering experience. LIFE-ALTERING, I say! Creamy, crunchy and flavorful, just like a real cannoli should be.

The only thing keeping me from granting Isgro's the coveted five star status are the prices. Ouch. A little steep for my liking, but I won't complain - it just might be the only thing keeping me from going there on a regular basis. And I am not prepared to buy a new wardrobe to support what my body-by-pastry would look like.
}

**items' history reviews text:**
{

Silk City is a diner by day and dance party by night. I've been here for brunch and lunch a few times both are very good. Defiantly ask about the specials before making your food choices. As for the dance party if you want to beat the cover get there just before 9 it'll be a little on the light side but the party gets moving pretty quickly.
}
{

I've heard of this place for a while. It's just fun. They do a good job with COVID and they have a lot of great drink and food options to be hand. The ambiance is eclectic - lots of mood lighting and plants. It kind of sets the mood of an oasis in the middle of a busy industrial neighborhood. Still, this is a staple in the Philadelphia community and you should definitely come here at least once. (PS. The drinks are prettyyy strong too...)
}
{

Looking for a place to get together with a friend on an unseasonably warm November day, I knew Silk City has an out door space (which is all that is now allowed in Philly during the pandemic) so I booked it for dinner. The outdoor garden is colorful and I read they also had heat lamps. When I arrived I was pleased to see the outdoor area looked as attractive as ever but I didn't feel totally comfortable as it was basically enclosed like a tent. I was hoping to see more vents and windows open for ventilation as it was a warm day. I'm not sure during this pandemic - what the difference is dining in a tent like structure which is permitted if it covers a previous outdoor area and dining inside in a large dining room which is forbidden. Anyway despite our reservations about it being totally safe, we decided to dine anyway. And Silk City came through with good food as usual.

A jalapeno-cheddar-corn muffin is the way all corn muffins should be made in my opinion. And my main course called 'Shaking Beef Bowl' was very flavorful with soba noodles, beef tenderloin, greens, and various veggies. I added a little cholula sauce and I totally got into it. My side order of Peruvian purple potatoes and veggies in a salsa verde was also good although I could have used a little more seasoning. A tasty strawberry cheese cake ended the meal.

Silk City is a long-running Philly institution. Though it has changed owners over the years, the quality of food remains high and the service excellent. And when you can park right in front of it, it's a great meet-up spot on the edge of Center City Philly.
}
{

Cozy diner with a classic American cuisine focus. The decor gives it a retro feel, kind of like stepping back in time. They have a huge menu with even bigger portions.

They have a bar you can sit and drink at.

I felt that their veggie chicken wings were slamming, and that they had solid sandwiches. Would definitely come back.
}
{

such a cool, retro place! eclectic vibe. really nice for weekend brunch or a night out dancing.
}

**Instruction prompt:** Below is the embedding for the review you need to predict: $<rev\_emb>$. Based on this embedding and any available historical context, please provid your prediction of the review.

---

**Result:**

First, let's analyze the buyer's historical reviews:

The buyer has written 5 reviews. The buyer's reviews are quite diverse in terms of the type of establishment they are reviewing (restaurant, diner, lounge, and bakery). The buyer seems to have a good sense of humor and often uses playful language in their reviews. They also seem to be a bit of a foodie, as they often mention the quality of the food and the presentation of the dishes. The buyer also seems to appreciate good service and a cozy atmosphere.

Now, let's analyze the historical reviews of the item (Silk City): The item's historical reviews are quite positive, with many reviewers mentioning the good food and service. The item seems to be a popular spot for both day and night, with some reviewers mentioning that it's a great place for brunch or a night out dancing. The item's reviews also mention the eclectic vibe and retro feel of the establishment.

Based on this information, I predict that the buyer's review of Silk City would be positive. The buyer seems to appreciate good food and service, and Silk City's historical reviews suggest that it excels in both areas. The buyer also seems to appreciate a cozy atmosphere, and Silk City's reviews mention the eclectic vibe and retro feel of the establishment. I predict that the buyer would mention the good food and service, as well as the cozy atmosphere, in their review. They may also mention the eclectic vibe and retro feel of the establishment, as well as the fact that it's a great place for both day and night.

Here's a possible review from the buyer:

"OMG, Silk City is a total gem! The food is amazing - I had the most incredible brunch there last weekend. The service was top-notch, and the atmosphere was so cozy and retro-chic. I loved the eclectic vibe and the fact that it's a great spot for both day and night. The prices were a bit steep, but totally worth it. I'm already planning my next visit!"

