# OpenReview forum: "Leveraging Historical Interactions for Factual Review Generation with Large Language Models"
_ICLR.cc/2026/Conference — ICLR 2026 Conference Withdrawn Submission_

### Official Review · Reviewer_LA6Q · 2025-10-19

**Soundness:** 3
**Presentation:** 4
**Contribution:** 3
**Rating:** 2
**Confidence:** 3

**Summary:**

This paper introduces DyGRevLLM, a framework for generating factually consistent user reviews using large language models (LLMs) by integrating dynamic graph representation learning. It models user-item interactions as a bipartite dynamic graph, pretrains embeddings to predict future review representations, and conditions LLMs on historical reviews and predicted embeddings to ensure factual accuracy, personalization, and coherence.

Key Contributions

- Framework Design: Combines dynamic graph learning (e.g., time-aware attention, GRU updates) with LLMs to incorporate evolving user-item interactions, addressing limitations in prior LLM-based methods.
- Pretraining Task: Predicts future review embeddings via MSE loss, aligning non-textual historical data with LLM semantics.
- Evaluation Metrics: Proposes Descriptive Recall Alignment (DRA) and Descriptive Jaccard Consistency (DJC) to measure factual consistency beyond BLEU/ROUGE.
- Empirical Validation: Outperforms baselines (e.g., PETER, ChatGPT-4o, Review-LLM) on Amazon and Yelp datasets in similarity and factual metrics.

**Strengths:**

Originality

The paper demonstrates strong originality through a creative integration of dynamic graph representation learning (e.g., time-aware attention and GRU-based state updates) with LLMs to address factual inconsistencies in review generation. This bridges non-textual, evolving user-item interactions into LLM semantics via a novel pretraining task that predicts future review embeddings aligned with LLM token spaces. While building on prior works like temporal graph networks (Rossi et al., 2020) and LLM-based review methods (Xie et al., 2023; Peng et al., 2024), it removes limitations by explicitly modeling temporal dynamics, which prior LLM approaches often overlook in favor of static prompts or fine-tuning.

Quality

The work exhibits high quality in its rigorous methodology and evaluation. The framework is technically sound, with well-formulated components (e.g., MSE loss for embedding prediction) and comprehensive experiments on real-world datasets (Amazon-Book, Amazon-Cloth, Yelp). It outperforms baselines across factual (DRA, DJC) and similarity metrics (BLEU, BERTScore), supported by ablations, sensitivity analyses, visualizations (e.g., t-SNE of embeddings), and human rankings. The proposed metrics are thoughtfully designed and validated against human judgments, enhancing reliability.

Clarity

The presentation is excellent, with a logical structure, precise writing, and effective visuals (e.g., Figure 2's architecture overview). Equations are clearly explained, and appendices provide detailed implementations, prompts, and examples, making the complex integration of graphs and LLMs accessible without ambiguity.

Significance

This contribution has notable significance for recommendation systems and e-commerce, where factually accurate reviews improve user trust and personalization. By mitigating LLM hallucinations through dynamic historical context, it advances practical applications in dynamic text generation. The new evaluation metrics (DRA, DJC) offer broader impact for assessing semantic alignment in LLM outputs, potentially extending to domains like sequential recommendations or time-series NLP.

**Weaknesses:**

Methodology

A key weakness lies in the integration of the predicted review representation $\hat{h}_r(t)$ into the LLM during the generation phase (Section 3.2). The paper states that $\hat{h}_r(t)$ is "embedded as a token <rev_emb> within the generation sequence," but it does not specify how this arbitrary embedding is handled by the off-the-shelf LLaMA3-8B model, which is not fine-tuned for such custom inputs. Since LLMs process tokenized inputs and their embedding layers are fixed, injecting a custom vector for a special token could lead to poor interpretation by the transformer's subsequent layers, potentially rendering the conditional guidance ineffective or noisy. This is evident in the lack of ablation studies isolating the impact of this embedding (e.g., comparing generation with vs. without it, or with random embeddings). To improve, the authors could fine-tune the LLM (or use adapters like LoRA) on a small set of examples where <rev_emb> is paired with target reviews, ensuring the model learns to condition on it meaningfully. Additionally, provide implementation details in the appendix on modifying the embedding matrix during inference, including code snippets for reproducibility.

Experiments

The experimental setup (Section 4) lacks sufficient ablation and sensitivity analysis to robustly validate the framework's components. For instance, while hyperparameters like $k=5$ for historical reviews are mentioned, there is no systematic evaluation of varying $k$ (e.g., 1, 3, 10) or its impact on users with sparse histories (common in Yelp, where some users may have <5 reviews). The baselines are unbalanced: open-source LLMs like ChatGLM-6B (6B parameters) are compared against LLaMA3-8B (8B), potentially inflating gains, and no fine-tuned LLM baselines (e.g., LLaMA fine-tuned on review data) are included to isolate the graph's contribution over simple prompt engineering. Human evaluation (implied in RQ1) appears limited, as Table 2 reports aggregated scores without details on annotator count, agreement (e.g., Cohen's kappa), or prompts used. To address this, expand ablations to include component-wise removal (e.g., without time-aware attention in Equation 6) and test on sparser datasets like MovieLens-1M for generalizability. Also, incorporate standard hallucination metrics like FactScore (Min et al., 2023) alongside DRA/DJC to benchmark factual improvements quantitatively.

Evaluation Metrics

The proposed metrics, Descriptive Recall Alignment (DRA) and Descriptive Jaccard Consistency (DJC) (Section 4.1), are under-specified in how "descriptive features" (e.g., adjectives, attributes) are extracted, relying on potentially brittle rule-based methods without validation against NLP tools like dependency parsing or entity recognition. This could bias results toward surface-level matches, as seen in similar custom metrics in review generation (e.g., in PRAG-LLM, Xie et al., 2023, which uses BERTScore for semantics). The paper mentions computation in Appendix A.3 but provides no correlation analysis with human judgments beyond overall performance. For improvement, automate feature extraction using pre-trained models like spaCy for adjectives/attributes, and conduct a metric validation study (e.g., Spearman correlation with 100+ human-annotated pairs) to ensure they capture factual alignment better than alternatives like Semantic Textual Similarity (STS). This would strengthen claims of overcoming BLEU/ROUGE limitations.

Scalability and Generalizability

The datasets (Amazon-Book, Amazon-Cloth, Yelp; Table 1) are relatively small (e.g., 4,825 users in Amazon-Book) and dated (spanning 4-11 years pre-2023), limiting insights into scalability for real-time systems with millions of interactions. The dynamic graph aggregation (Equation 3) and GRU updates (Equation 4) could become computationally expensive on larger graphs, but no runtime analysis or efficiency metrics (e.g., FLOPs per user) are provided. Generalizability is further constrained by English-only reviews and e-commerce focus, ignoring multilingual or domain shifts (e.g., social media reviews). To mitigate, test on larger, recent datasets like Amazon-2023 or Steam reviews, report inference times, and explore optimizations like graph sampling (e.g., via PinSage) to handle scale, ensuring the framework meets its goal of practical deployment in dynamic recommendation systems.

**Questions:**

Embedding Integration in LLM Generation: How is the predicted review embedding $\hat{h}_r(t)$ technically injected as the <rev_emb> token into an off-the-shelf LLaMA3-8B without fine-tuning, given fixed embedding layers? An ablation comparing generation with/without it or with random embeddings could clarify its impact; if ineffective, consider adapters (e.g., LoRA) for better conditioning, potentially changing my view on methodological soundness.

Ablations and Sensitivity Analysis: Why no evaluation of varying historical review length $k$ (e.g., 1-10) or performance on sparse users (common in Yelp)? Adding these, plus component ablations (e.g., without time-aware attention), could address generalizability; results might resolve concerns about robustness and alter my assessment of experimental quality.

Baseline Balance and Human Evaluation: Why compare smaller open-source LLMs (e.g., ChatGLM-6B) to LLaMA3-8B, and no fine-tuned LLM baselines? Provide details on human eval (e.g., annotator count, Cohen's kappa). Including these could strengthen claims of superiority

---

### Official Review · Reviewer_PpAB · 2025-10-30

**Soundness:** 3
**Presentation:** 3
**Contribution:** 3
**Rating:** 6
**Confidence:** 2

**Summary:**

This paper tackles factual review generation using LLMs, addressing the challenge that existing LLM-based methods often fail to incorporate dynamic and personalized user–item interaction histories. The authors propose DyGRevLLM, which integrates dynamic graph representation learning with LLM prompting. A pretraining phase constructs future review representations by modeling temporal interaction patterns, and these representations are used as conditional signals during generation. Experiments across Amazon and Yelp datasets demonstrate improved factual consistency and relevance compared to classical neural models, standalone LLMs, and recent LLM-based review generators. Two new factual evaluation metrics (DRA, DJC) are also introduced and correlated with human judgment.

**Strengths:**

1. Motivation: The paper clearly identifies a meaningful problem in existing LLM-based review generation—although LLMs generate fluent text, they often fail to incorporate dynamic user–item interaction histories, resulting in reviews that lack personalization and factual grounding. This limitation is real and practically relevant, and the paper provides clear reasoning on why simple prompting or retrieval cannot fully address the issue.
2. Challenge: The challenges described are substantial and rooted in the nature of the task rather than artificially constructed. The paper targets (1) modeling evolving user/item preferences, (2) mapping non-text temporal interaction signals into LLM semantic space, and (3) evaluating factual consistency. These challenges are non-trivial and are articulated clearly, demonstrating meaningful difficulty in the task.

**Weaknesses:**

1. Technical Soundness: While the overall model pipeline is logically coherent, the paper does not sufficiently explain how the predicted future review representation (<rev_emb>) influences LLM generation behavior. There is no analysis or visualization of how the embedding affects token probabilities or semantic direction. As a result, this key mechanism functions as a black box, which weakens the interpretability and verifiability of the proposed approach.
2. Experiments: Although baseline comparisons and ablations are presented, the experimental section lacks efficiency and scalability analysis. The paper does not discuss training cost or memory footprint of dynamic graph modeling, nor does it evaluate the method across different LLM backbone sizes. Without complexity comparison or scalability experiments, it remains unclear whether the method can be deployed or scaled in large real-world systems.

**Questions:**

1. Could the authors provide an explicit example of the full prompt format including <rev_emb>?
2. How does model performance change when scaling to larger LLM backbones (e.g., 13B, 70B)?
3. How sensitive is the method to the length and recency of historical interactions beyond the tested k=5?
4. Can the predicted representations be interpreted (e.g., clustering of semantic dimensions)?

**Details Of Ethics Concerns:**

No ethical issues identified.

---

### Official Review · Reviewer_G91K · 2025-10-30

**Soundness:** 2
**Presentation:** 3
**Contribution:** 2
**Rating:** 4
**Confidence:** 3

**Summary:**

This paper addresses the issue of “factual inconsistencies” that large language models may generate when producing product or service reviews, which can deviate from users' genuine preferences or the actual attributes of items. To tackle this challenge, the authors propose an innovative framework named DyGRevLLM. The core concept of this framework lies in integrating dynamic graph representation learning with the generative capabilities of large language models. The paper also introduces novel evaluation metrics (DRA and DJC) specifically designed to measure factual consistency, demonstrating the method's superiority across multiple real-world datasets.

**Strengths:**

1. The paper points out that existing large language models (LLMs) cannot understand and utilize non-textual, dynamically evolving user-item interaction signals. It breaks down this issue into three challenges: dynamic signal capture, modality alignment, and factual evaluation. It proposes a method that integrates dynamic graphs with large models to mitigate this problem.

2. The paper compares traditional deep learning models, open-source/closed-source general-purpose LLMs, and the latest LLM review generation methods. Experiments demonstrate the model's performance advantages.

3. The paper is easy to follow.

**Weaknesses:**

1.  Although the paper introduces two new metrics, DRA and DJC, it fails to provide sufficient evidence demonstrating their ability to accurately measure “factuality.” These metrics essentially remain semantic comparisons against “target comments,” and may prove ineffective if the target comments themselves are ambiguous or non-descriptive.

2. The core interactive signal for the model is the “commenting behavior” itself. However, in real-world recommendation systems (such as Amazon), there also exist various implicit feedback signals including clicks, adding to cart, ratings, and browsing duration. These rich interactive signals, which may precede comments, are often overlooked. This oversight leads to the model capturing potentially incomplete user intent and item characteristics.

3. The article fails to clearly demonstrate that the performance improvement genuinely stems from the “dynamic evolutionary information” captured by the dynamic diagram, rather than simply from providing the LLM with richer, more structured contextual information.

4. The representations learned by dynamic images and the word embedding space of LLMs seem to exhibit fundamental differences in geometric structure and semantic distribution. Forcing alignment through regression loss may result in information loss.

**Questions:**

1. How ensure that simple numerical regression produces semantically valid alignments, rather than turning the <rev_emb> token into “noise” that the LLM cannot interpret correctly?

2. The paper identifies LLM's inability to comprehend dynamic interaction signal sequences as a core challenge. But is this fundamentally a data input issue rather than a capability issue? If an LLM is pre-trained on user-item interaction data with temporal tags, could it autonomously learn these dynamic patterns from raw text? Does this approach merely compensate for the shortcomings of current LLM pre-training data?

---

### Note · Authors · 2025-11-16

**Comment:**

We sincerely thank the reviewers and the area chair for their time and constructive feedback on our submission. After careful internal discussion, we have decided to withdraw this submission from ICLR. We greatly appreciate the comments provided in the reviews and will carefully take them into account when revising and improving the manuscript.

**Withdrawal Confirmation:**

I have read and agree with the venue's withdrawal policy on behalf of myself and my co-authors.